# A DEM-Based Modeling Method and Simulation Parameter Selection for *Cyperus esculentus* Seeds

Tianyue Xu [1], Ruxin Zhang [1], Fengwu Zhu [1], Weizhi Feng [1], Yang Wang [2] and Jingli Wang [1,*]

1    College of Engineering and Technology, Jilin Agricultural University, Changchun 130118, China
2    College of Biological and Agricultural Engineering, Jilin University, Changchun 130021, China
*    Correspondence: wjlwy2004@sina.com; Tel.: +86-13500819883

**Abstract:** To build a DEM model of *Cyperus esculentus* seed particles, the shape and size of the *Cyperus esculentus* seed particles were measured and analyzed. The results showed that the dispersity in size had a normal distribution. Additionally, a certain functional relationship between the primary dimension and secondary dimensions was determined. The width of the seed was the primary dimension, and the other secondary dimensions (length and thickness) were calculated based on their relationships with the primary dimension. On this basis, an approach for modeling *Cyperus esculentus* seed particles based on the multi-sphere (MS) method was proposed. The discrete element analysis models of three varieties of *Cyperus esculentus* seeds were established with different numbers of filing spheres. Moreover, to obtain more accurate simulation parameters, first, a range of values of the simulation parameters was obtained by the experimental method. Second, the Plackett–Burman (PB) test and the path of steepest ascent method were both adopted to correct and calibrate the simulation parameters, which were difficult to obtain through experiments, and simulation of the direct shear test was used for calibration. All of the methods guaranteed that the selected parameters were reasonable. The test results showed that the static friction coefficient of seed–seed had a significant effect on the simulation results. Finally, piling tests and the bulk density test were used for modeling verification. By comparing the simulated results and experimental results in the piling tests and bulk density test, when the number of filing spheres increased, the simulated results were close to those obtained experimentally. Therefore, the feasibility and validity of the modeling method for *Cyperus esculentus* seed particles that we proposed and the simulation parameters that were obtained were verified.

**Keywords:** discrete element method; *Cyperus esculentus*; parameter selection; simulation; modeling

## 1. Introduction

*Cyperus esculentus* is a type of cash crop from which oil, grain, feed, and medicine can be derived, and the cultivated area is increasing year by year [1]. However, the mechanization level for processing *Cyperus esculentus* is not adequate for needs, so it is necessary to urgently develop the related machinery for *Cyperus esculentus*. When *Cyperus esculentus* seeds or kernels are handled in seeding, harvesting, threshing, separation, processing, and packing, contacts occur between the individual *Cyperus esculentus* particles, as well as between the *Cyperus esculentus* particles and the related working components [2,3]. To analyze these contacts and therefore optimize the relevant working components, it is essential to build a precise model of *Cyperus esculentus* seed or kernel assembly, using the discrete element method [4].

The discrete element method (DEM) was proposed by Cundall et al. in 1971, and has been extensively used to calibrate the contact parameters between the particles and mechanical components. The method can be used to predict the mechanical behavior and motion of the particles. It is widely used for various industries, such as geomechanics, mining, pharmacy, agriculture, food, and the chemical industry [4]. Accurately representing the shape of the particles is the key to DEM analysis. Compared with those of soybean,

corn, and wheat seeds, the shape of *Cyperus esculentus* seeds is more irregular. When modeling irregular seed particles, they can be represented by ellipsoids [5,6], superquadrics [7], and polyhedrons [8]. However, the contact detection algorithms for both ellipsoids and superquadrics are complicated, which renders them time-consuming [9]. Currently, many researchers have used the multi-sphere (MS) method to build non-spherical particle models [10]. Xu et al. [11] and Yan et al. [12], respectively, established 5-sphere, 9-sphere, and 13-sphere models for soybean seeds with different sphericities by using the MS method. Chen et al. established 10-sphere to 14-sphere, 18-sphere, and 6-sphere models for corn seeds with three shapes of horse-tooth, cone, and sphere [13]. Zhou et al. proposed a general modeling method that was applicable to corn seed particles [14]. However, it is still necessary to further analyze which method can be adopted to establish a general modeling method for *Cyperus esculentus*.

In addition, when DEM is adopted for simulation, most scholars find that the simulation parameter has a great influence on the simulation results through the analysis of parameter sensitivity [15]. Some of the parameters, such as the elastic modulus and restitution coefficient, can be obtained by experiments, but the friction coefficient is difficult to obtain directly by tests, so it needs to be calibrated. Wang et al. used the golden section method combined with a single-factor experiment to determine the rolling friction coefficient of corn particles [16]. The simulation parameters of Fagopyrum esculentum were calibrated through a piling test combined with the PB test by Fan et al. [17]. Therefore, how to select and identify the simulation parameters by means of experiments and calibrations should be further studied.

Meanwhile, discrete element simulation technology greatly depends on the established model, and the modeling accuracy directly determines the reliability of the simulation results. There exist some shortcomings in DEM modeling of irregular agricultural materials, and few studies have been conducted on the simulation parameters and optimization of irregular seed particle models.

Therefore, in this paper, three varieties of *Cyperus esculentus* seeds were used to measure and analyze their shapes and sizes. The proposed modeling methods for a single *Cyperus esculentus* seed particle and *Cyperus esculentus* seed assembly were based on the multi-sphere method. The physical and mechanical properties of the *Cyperus esculentus* seed, such as density, moisture content, restitution coefficient, elastic modulus, and static friction coefficient, were measured and analyzed. Furthermore, to clarify the influence of the simulation parameters on the simulation results, by means of simulation of the direct shear test, the PB test combined with the path of steepest ascent method was adopted to select the simulation parameters with a significant performance impact. Then, calibration was achieved to guarantee that the parameter selection was reasonable. Finally, by comparing the simulated and experimental results in piling tests and the bulk density test, the feasibility and validity of the modeling methods for the *Cyperus esculentus* seed particles and model parameter selection were verified. This work will provide a reference for related research.

## 2. Measurement and Analysis of the Physical Properties of *Cyperus esculentus*

In this paper, three varieties (Jinong 1, Jinong 2, and Jinong 3) of *Cyperus esculentus* seeds were used, as shown in Figure 1. An electronic balance with an accuracy of 0.01 g was used to measure the thousand seed weight of the seeds, and the moisture content of the seeds was measured by the oven drying method. In addition, the density of the seeds was measured by the liquid replacement method, and the experimental results are listed in Table 1.

**Table 1.** Density, moisture content and thousand seed weight of the *Cyperus esculentus* seeds.

| Variety | Density | Moisture Content | Thousand Seed Weight |
|---------|---------|------------------|----------------------|
| Jinong 1 | 1.34 g/cm$^3$ | 28.8% | 427 g |
| Jinong 2 | 1.27 g/cm$^3$ | 28.4% | 406 g |
| Jinong 3 | 1.19 g/cm$^3$ | 35.8% | 809 g |

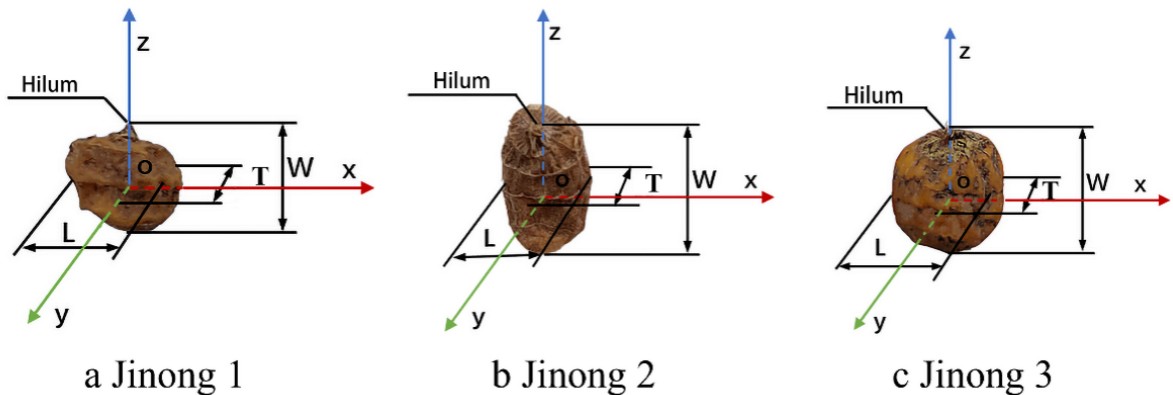

**Figure 1.** Tri-axial dimensions and reference coordinate system of the *Cyperus esculentus* seeds.

　　The tri-axial dimensions of the three varieties of *Cyperus esculentus* seeds were defined as the length (*L*), width (*W*), and thickness (*T*), as shown in Figure 1a–c. One hundred intact seeds were randomly selected from each variety, and the sizes were measured by a digital Vernier caliper with an accuracy of 0.01 mm. The mean value and standard deviation of the tri-axial dimensions of the *Cyperus esculentus* seeds are listed in Table 2.

**Table 2.** Size of the different varieties of *Cyperus esculentus* seeds.

| Variety | Size | Mean/mm | Standard Deviation/mm |
|---------|------|---------|----------------------|
| Jinong 1 | Length (*L*) | 9.63 | 0.54 |
|  | Width (*W*) | 9.11 | 1.19 |
|  | Thickness (*T*) | 7.94 | 1.43 |
| Jinong 2 | Length (*L*) | 8.02 | 0.62 |
|  | Width (*W*) | 13.70 | 1.71 |
|  | Thickness (*T*) | 5.80 | 0.65 |
| Jinong 3 | Length (*L*) | 11.86 | 1.37 |
|  | Width (*W*) | 11.45 | 1.56 |
|  | Thickness (*T*) | 9.49 | 1.56 |

　　The results showed that the sizes all followed a normal distribution. The distributions of the tri-axial dimensions are shown in Figures 2–4.

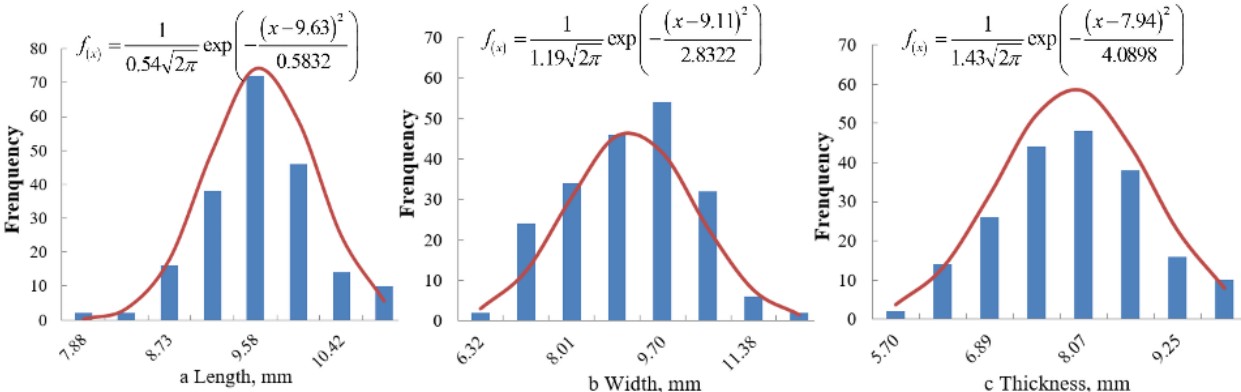

**Figure 2.** Distribution of the size of Jinong 1 seeds: (**a**) length; (**b**) width; and (**c**) thickness.

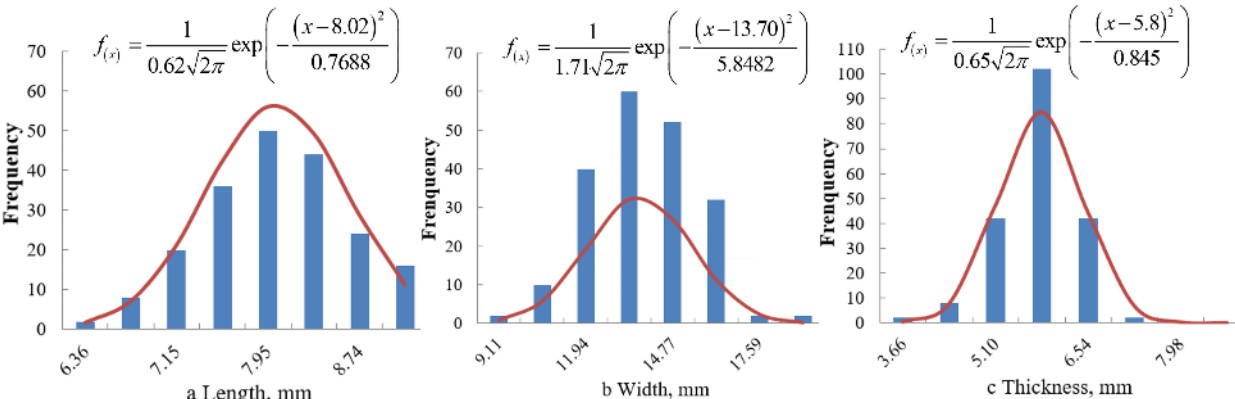

**Figure 3.** Distribution of the size of Jinong 2 seeds: (**a**) length; (**b**) width; and (**c**) thickness.

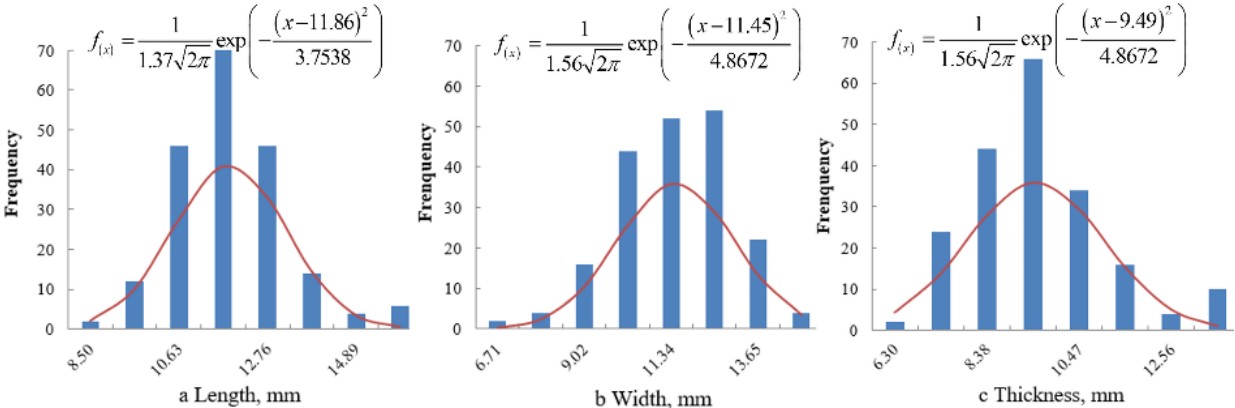

**Figure 4.** Distribution of the size of Jinong 3 seeds: (**a**) length; (**b**) width; and (**c**) thickness.

The tri-axial dimensions of the *Cyperus esculentus* seeds were analyzed, and functional relationships were found between the width–length and width–thickness of the *Cyperus esculentus* seeds, as shown in Figures 5–7. According to the analysis, the width was defined as the primary dimension, while the length and thickness were the secondary dimensions. The relationships between the primary dimension and the secondary dimensions of the seeds of the three varieties of *Cyperus esculentus* are listed in Table 3, and the lengths and thicknesses were calculated according to the expressions.

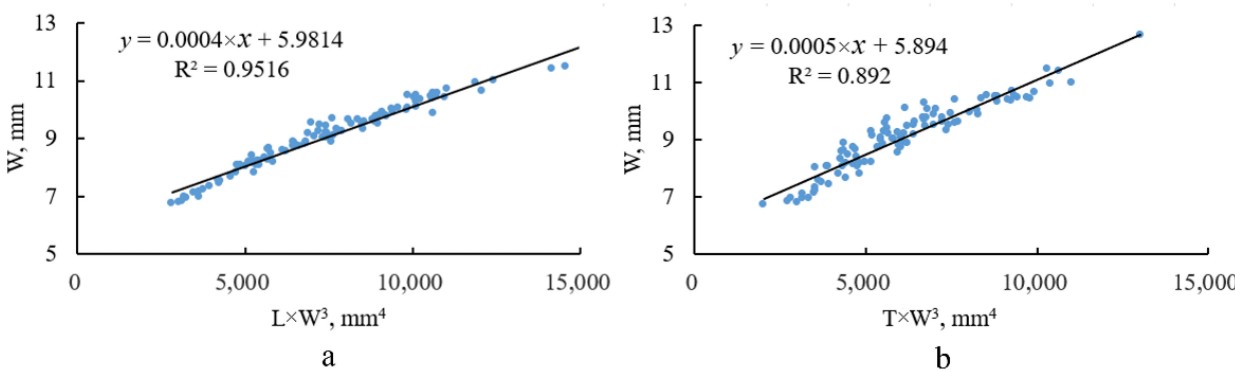

**Figure 5.** Relationships between two dimensions of Jinong 1 seeds: (**a**) width–length and (**b**) width–thickness.

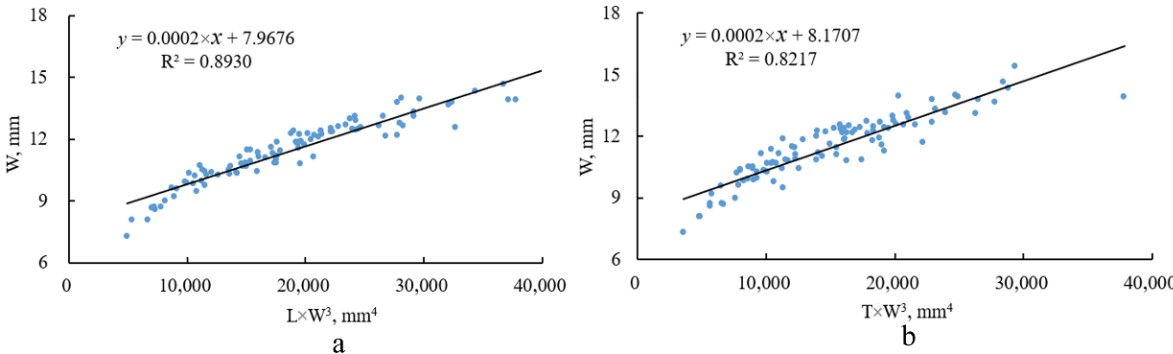

**Figure 6.** Relationships between two dimensions of Jinong 2 seeds: (**a**) width–length and (**b**) width–thickness.

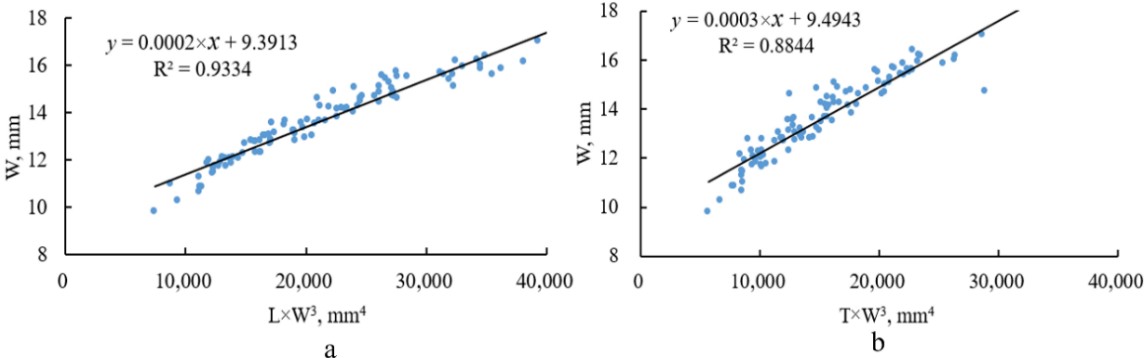

**Figure 7.** Relationships between two dimensions of Jinong 3 seeds: (**a**) width–length and (**b**) width–thickness.

**Table 3.** Relationships between the primary dimension and the secondary dimensions of different varieties of *Cyperus esculentus* seeds.

| Variety | Expression | $R^2$ |
|---|---|---|
| Jinong 1 | $L = (W - 5.9814)/(0.0004 \times W^3)$ <br> $T = (W - 5.984)/(0.0005 \times W^3)$ | 0.9516 <br> 0.892 |
| Jinong 2 | $L = (W - 9.3913)/(0.0002 \times W^3)$ <br> $T = (W - 9.4943)/(0.0003 \times W^3)$ | 0.9334 <br> 0.8844 |
| Jinong 3 | $L = (W - 7.9676)/(0.0002 \times W^3)$ <br> $T = (W - 8.1707)/(0.0002 \times W^3)$ | 0.893 <br> 0.8217 |

In summary, when modeling the *Cyperus esculentus* seed assembly, the width (*W*) was selected as the primary dimension and was generated according to the normal distribution, and the other two dimensions (*L* and *T*) were calculated according to their relationships with the primary dimension. As a result, the size and distribution of the created seed assembly were close to those of the actual *Cyperus esculentus* seed assembly.

## 3. Measurement and Analysis of the Mechanical Properties of *Cyperus esculentus*

The mechanical properties of the three varieties of *Cyperus esculentus* seeds were measured and analyzed. The elastic modulus is a measurement of a material's resistance to elastic deformation [18]. In this paper, the elastic modulus of the *Cyperus esculentus* seeds was measured by a compression test using a universal testing machine, as shown in Figure 8. The curvature radius between the seed–upper contact surface and the seed–lower

contact surface was considered to be the same, and the elastic modulus was calculated according to the simplified Hertz formula in Equation (1):

$$E^* = \frac{0.338F\left(1 - \mu^2\right)}{D^{\frac{3}{2}}} \left[2K_U\left(\frac{1}{R} + \frac{1}{R'}\right)^{\frac{1}{3}}\right]^{\frac{3}{2}} \tag{1}$$

where $E^*$ is the elastic modulus of the seed, Pa; $F$ is the force loaded normally to the seed, N; $D$ is the deformation of the seed, mm; $\mu$ is the Poisson's ratio of the seed; $R$ and $R'$ are the curvature radii of the seeds when in contact with the compression probe or undersurface, mm; and $Ku$ is a constant, which can be determined according to the relationship between $\cos\theta$ and $Ku$ by looking up a table (the standard of the ASAE S368.4 Dec2000 (R2017) [19]), where the value of $\cos\theta$ is calculated by Equation (2):

$$\cos\theta = \left[\frac{1}{R} - \frac{1}{R'}\right] \Big/ \left[\frac{1}{R} + \frac{1}{R'}\right] \tag{2}$$

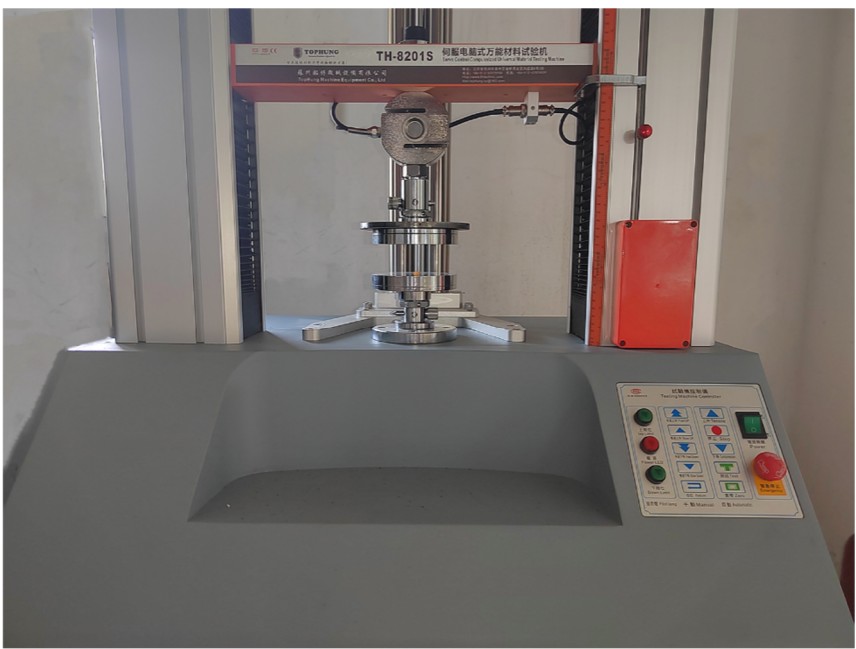

**Figure 8.** Compression test by a universal testing machine.

The curvature radii of $R$ and $R'$ were calculated according to the empirical formulas in Equations (3) and (4):

$$R = \frac{H'}{2} \tag{3}$$

$$R' = \frac{H^2 + \frac{L^2}{4}}{2H} \tag{4}$$

where $H'$ is the thickness of the seed when compressed, mm; and $L'$ is the length of the seed when compressed, mm.

Due to the irregular shape of the seeds and the current experimental conditions, it is difficult to measure Poisson's ratio accurately. Therefore, a value of 0.4 is adopted which is suitable for general beans [19]. Thus, the elasticity moduli of the three varieties of *Cyperus esculentus* seeds were obtained, and the results for Jinong 1, Jinong 2, and Jinong 3 were $3.53 \times 10^7$ Pa, $5.55 \times 10^7$ Pa, and $1.46 \times 10^8$ Pa, respectively.

The shear modulus of the three varieties of *Cyperus esculentus* seeds was calculated based on the value of elastic modulus that we obtained previously, following Equation (5):

$$G^* = \frac{E^*}{2(1+\mu)} \tag{5}$$

where $G^*$ is the shear modulus of the seed, Pa; $E^*$ is the elastic modulus of the seed, Pa; and $\mu$ is the Poisson's ratio of the seed.

The restitution coefficients of seed–copper, seed–steel and seed–polymethyl methacrylate were measured and analyzed using a free fall test [20], as shown in Figure 9a. The restitution coefficient is the ratio of the descent velocity to rebound velocity, and the expression of the restitution coefficient was deduced combined with the kinematics equation, as shown in Equation (6):

$$e^* = \sqrt{\frac{h}{H}} \tag{6}$$

where $H$ is the release height of the seed, mm; and $h$ is the rebound height of the seed, mm.

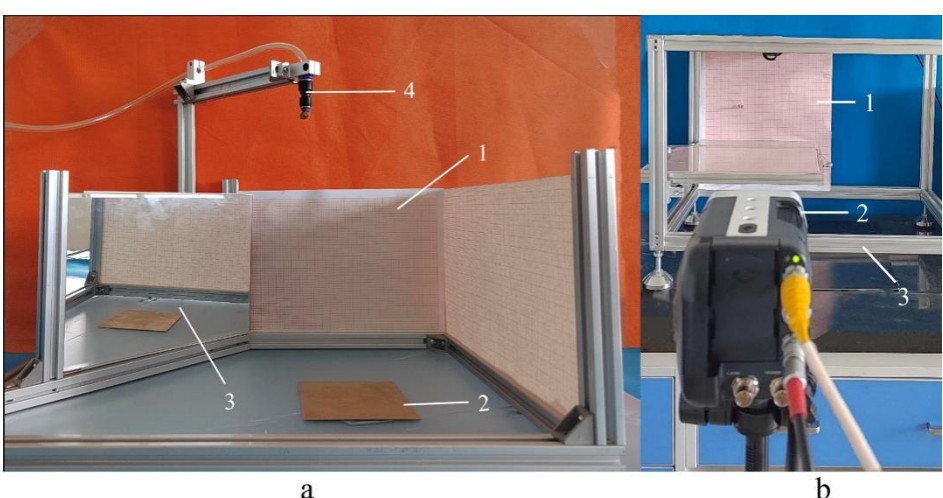

a         b

**Figure 9.** a–b Collision experiments: (**a**) Free fall test, 1—coordinate paper, 2—copper plate, 3—mirror, 4—air pump nozzle; (**b**) Single pendulum impact test, 1—coordinate paper, 2—high-speed camera, 3—self-made test bench.

The seed–seed restitution coefficient was measured by a single pendulum impact test [21], as shown in Figure 9b. The restitution coefficient between seed–seed was calculated following Equation (7):

$$e^* = \frac{v_1 - v_2}{v_0} = \frac{\sqrt{2gh_1} - \sqrt{2gh_2}}{\sqrt{2gH}} \tag{7}$$

where $v_0$ is the release velocity of No. 1 seed, m/s; $v_1$ is the velocity of No. 2 seed (collided seed) after impact, m/s; $v_2$ is the velocity of No. 1 seed after impact, m/s; $H$ is the release height of No. 1 seed, mm; $h_1$ is the rebound height of No. 2 seed (collided seed), mm; and $h_2$ is the rebound height of No. 1 seed, mm.

The restitution coefficient between the *Cyperus esculentus* seed particles and the restitution coefficient of seed–copper, seed–steel, and seed–polymethyl methacrylate were measured, and the results are listed in Table 4.

**Table 4.** Restitution coefficient of different varieties of *Cyperus esculentus* seeds.

| Variety | Collision Material | Restitution Coefficient |
|---|---|---|
| Jinong 1 | Copper | 0.58 |
| | Steel | 0.65 |
| | Polymethyl methacrylate | 0.41 |
| | Seed | 0.28 |
| Jinong 2 | Copper | 0.64 |
| | Steel | 0.75 |
| | Polymethyl methacrylate | 0.42 |
| | Seed | 0.34 |
| Jinong 3 | Copper | 0.68 |
| | Steel | 0.79 |
| | Polymethyl methacrylate | 0.55 |
| | Seed | 0.50 |

Moreover, the static friction coefficients of seed–copper, seed–steel, and seed–polymethyl methacrylate were measured and analyzed. When the static friction between the seed and contact material occurs, the seed should have a tendency to slide. In this paper, the slope method [22] was used for measurement with a home-made, inclined apparatus and angle sensor, as shown in Figure 10a. To prevent the seeds from rolling, nine seeds were stuck to a plate, as shown in Figure 10b. The static friction coefficients between the seeds and the different contact materials were calculated based on the mechanical formula in Equation (8), and the experimental results are listed in Table 5.

$$mgsin\theta = \mu_x mgcos\theta \tag{8}$$

where $\mu_x$ is the static friction coefficient; and $\theta$ is the angle between the inclined apparatus and horizontal plane, deg.

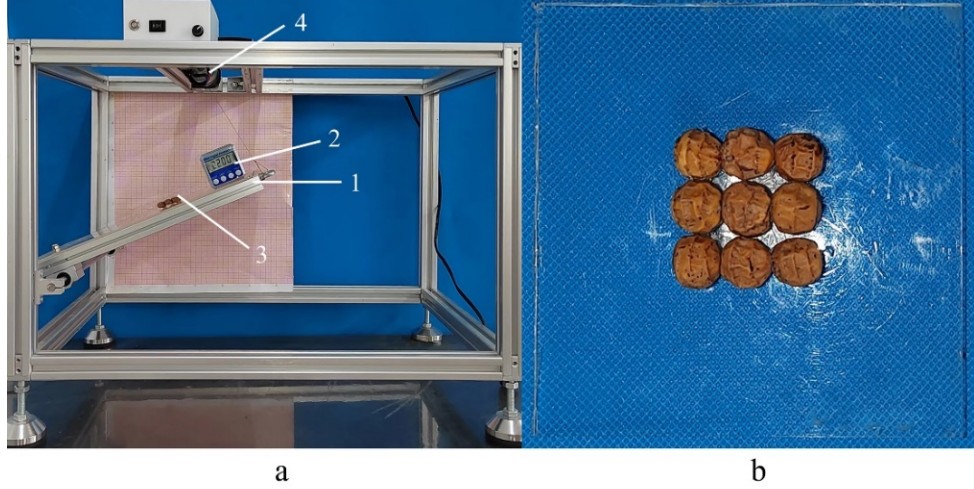

**Figure 10.** (**a**,**b**) Static friction coefficient measurement tests: (**a**) Measurement device: 1—inclinometer, 2—angle transducer, 3—seed plate, 4—electromotor; (**b**) Seed plate.

**Table 5.** Static friction coefficient for different varieties of *Cyperus esculentus* seeds.

| Material | Jinong 1 | Jinong 2 | Jinong 3 |
|---|---|---|---|
| Copper | 0.41 | 0.39 | 0.38 |
| Steel | 0.40 | 0.39 | 0.35 |
| Polymethyl methacrylate | 0.42 | 0.40 | 0.34 |

When rolling friction between the seed and contact material occurs, the motion of the seed should be self-rotation while rolling on the material. Thus, it is difficult to obtain the rolling friction coefficient between the seed and contact material. Moreover, the rolling friction coefficient between seed–seed was obtained by calibration; see the details in Section 5.

### 4. Modeling Method of *Cyperus esculentus* Seeds

#### 4.1. Particle Modeling

In this paper, the point cloud data of the outlines of three varieties of *Cyperus esculentus* seeds were obtained using a Minolta Vivid 910 3D laser scanner (Minolta Co., Osaka, Japan) with an accuracy of 0.05 mm. Based on the point cloud data of the outlines of the *Cyperus esculentus* seed, a single seed particle was modeled by the MS method. When filing spheres, the hilum of the seed was separately considered as a single sphere. The other parts of the seed were simplified as ellipsoids, which were the same as the actual parts with length ($L$), width ($W$), and thickness ($T$).

The principle of filing spheres was as follows: the outlines of the filing sphere were tangential to the outlines of the ellipsoid as far as possible; the center of the filing sphere was on the axis or axis plane of the ellipsoid; to meet the requirements for the filing accuracy, the number of filing spheres should be as small as possible.

The filing method for the 7-sphere model of Jinong 1 was as follows: the length ($L$), width ($W$), and thickness ($T$) of the seed were aligned on the major axis (axis $x$), the middle axis (axis $z$), and the minor axis (axis $y$) of the ellipsoid, respectively. On the $xoy$ plane, first, the maximum sphere (sphere $O_1$) with a radius of $T/2$ was filled at the center (point $O_1$) of the ellipsoid, as shown in Figure 11a. Second, on the $xoz$ plane, the width $W$ of the ellipsoid was divided into trisections, and two line segments, $\overline{AB}$ and $\overline{CD}$, which were parallel to axis $x$ and intersecting with the ellipsoid outline, were made through each equal diversion point. Then, the parallel line segments $\overline{AB}$ and $\overline{CD}$ were divided into trisections, as shown in Figure 11b. The equal diversion points (point $O_2$, point $O_3$, point $O_4$, point $O_5$, and point $O_6$) were the centers of the new filing spheres. Thus, another five spheres that were tangential to the outline of the ellipsoid were filled, defined as sphere $O_2$, sphere $O_3$, sphere $O_4$, sphere $O_5$, and sphere $O_6$, as shown in Figure 11c. Finally, the last sphere $O_7$ was filled at the position of the hilum. As a result, seven total spheres were filled. Based on the filing principle, the 7-sphere, 9-sphere, and 11-sphere models of Jinong 1 were successively modeled, as shown in Figure 14a–c.

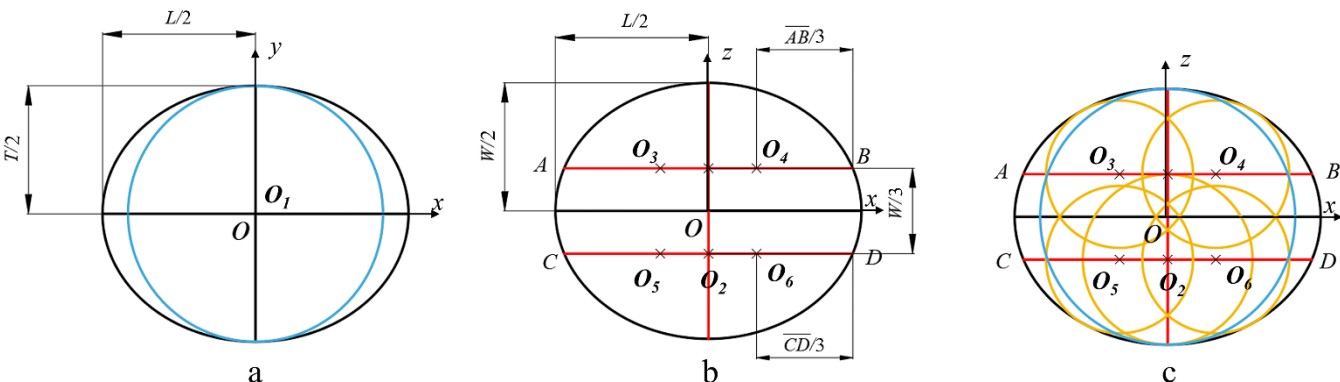

**Figure 11.** (**a**–**c**) Filing methods for the multi-sphere model of Jinong 1.

The filing method for the 9-sphere model of Jinong 2 was as follows: the width ($W$), length ($L$), and thickness ($T$) of the seed were aligned on the major axis (axis $x$), the middle axis (axis $z$), and the minor axis (axis $y$) of the ellipsoid, respectively. On the $xoz$ plane, first, the width $W$ of the ellipsoid was divided into four segments. Two line segments $\overline{AB}$ and $\overline{CD}$, which were parallel to axis $z$ and intersecting with the ellipsoid outline, were made through only two diversion points. Then, the parallel line segments $\overline{AB}$ and $\overline{CD}$ were divided into trisections, as shown in Figure 12a. Moreover, the equal diversion points

(point $O_1$, point $O_2$, point $O_3$, point $O_4$, point $O_5$, and point $O_6$) were the centers of the filing spheres. Thus, six spheres that were tangential to the outline of the ellipsoid were filled, defined as sphere $O_1$, sphere $O_2$, sphere $O_3$, sphere $O_4$, sphere $O_5$, and sphere $O_6$, as shown in Figure 12b. In addition, on the *xoy* plane, the width *W* of the ellipsoid was divided into seven parts, and the equal diversion points (point $O_7$ and point $O_8$) were only considered as the centers of the new filing spheres. Thus, another two spheres that were tangential to the outline of the ellipsoid were filled, defined as sphere $O_7$ and sphere $O_8$, as shown in Figure 12c. Finally, the last sphere $O_9$ was filled at the position of the hilum. As a result, nine spheres in total were filled. Based on the filing principle, the 9-sphere, 11-sphere, and 13-sphere models of Jinong 2 were successively modeled, as shown in Figure 14d–f.

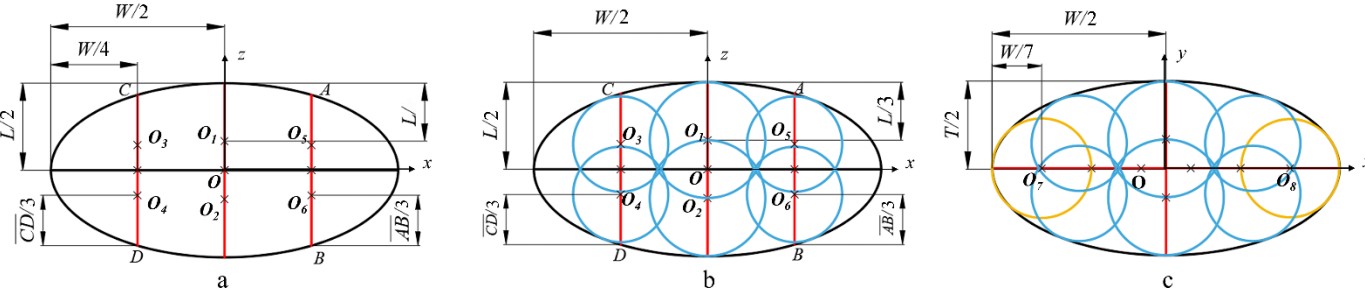

**Figure 12.** (**a**–**c**) Filing methods for the multi-sphere model of Jinong 2.

The filing method for the 7-sphere model of Jinong 3 was as follows: the width (*W*), length (*L*), and thickness (*T*) of the seed were aligned on the major axis (axis *x*), the middle axis (axis *z*), and the minor axis (axis *y*) of the ellipsoid, respectively. On the *xoz* plane, first, the width *W* of the ellipsoid was divided into trisections, and two line segments $\overline{AB}$ and $\overline{CD}$, which were parallel to axis *z* and intersecting with the ellipsoid outline, were made through the diversion points. Then, the parallel line segments $\overline{AB}$ and $\overline{CD}$ were divided into trisections, as shown in Figure 13a. Moreover, the equal diversion points (point $O_1$, point $O_2$, point $O_3$, and point $O_4$) were the centers of the new filing spheres. Thus, four spheres that were tangential to the outline of the ellipsoid were filled, defined as sphere $O_1$, sphere $O_2$, sphere $O_3$, and sphere $O_4$, as shown in Figure 13b.

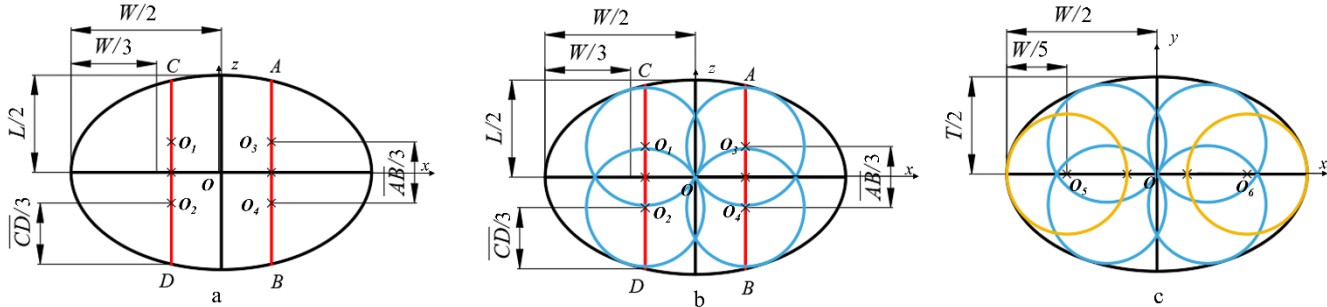

**Figure 13.** (**a**–**c**) Filing methods for the multi-sphere model of Jinong 3.

In addition, on the *xoy* plane, the width *W* of the ellipsoid was divided into five parts, and the equal diversion points (point $O_5$ and point $O_6$) were only considered as the centers of the new filing spheres. Thus, another two spheres that were tangential to the outline of the ellipsoid were filled, defined as sphere $O_5$ and sphere $O_6$, as shown in Figure 13c. Finally, the last sphere $O_7$ was filled at the position of the hilum. As a result, seven total spheres were filled. Based on the filing principle, 7-sphere, 9-sphere, and 11-sphere models of Jinong 3 were successively modeled, as shown in Figure 14g–i.

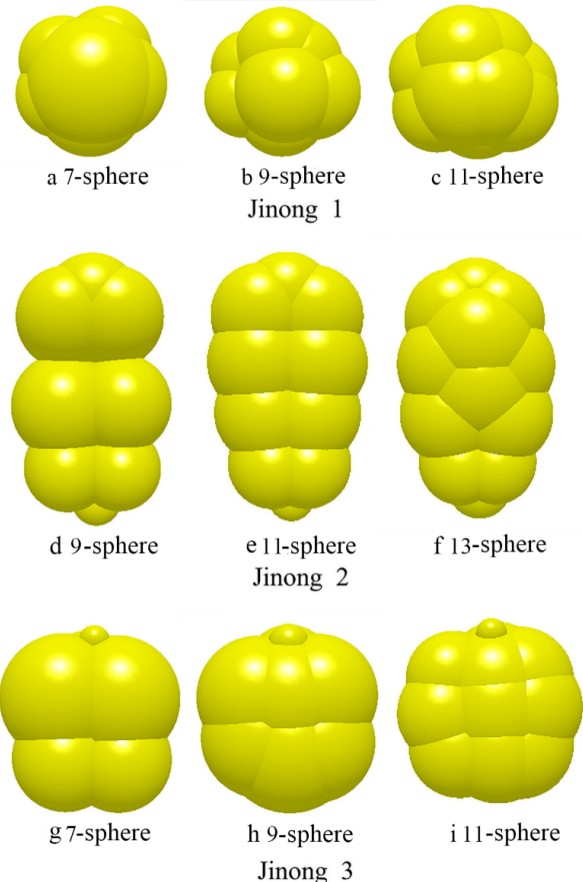

**Figure 14.** (**a–i**) Multisphere models for different varieties of *Cyperus esculentus* seeds.

### 4.2. Contact Force Model of Multi-Sphere Particles

As a partial sphere *a* in a multi-sphere particle *z*, a normal force and a tangential force which were acting on partial sphere *a* at the contact point *p*, given as:

$$\text{F}^n_{zap} = \left( \frac{4}{3}E^* \sqrt{R^*}\delta_{zap}^{3/2} - 2\sqrt{\frac{5}{6}}\beta\sqrt{S_n m^*}\text{v}^n_{zap} \right) \hat{\text{n}}_{zap} \tag{9}$$

and

$$\text{F}^t_{zap} = -\min\left( S_t \xi_{zap} + 2\sqrt{\frac{5}{6}}\beta\sqrt{S_t m^*}\text{v}^t_{zap}, \mu_s |\text{F}x| \frac{\xi_{zap}}{\left|\xi_{zap}\right|} \right) \tag{10}$$

where $E^*$ as the equivalent Young's modulus, and $E^* = \left[ \left(1 - \mu_z^2\right)/E_z + \left(1 - \mu_y^2\right)/E_y \right]^{-1}$ with $E_z$, $E_y$, $\mu_z$, and $\mu_y$ being the Young's moduli and Poisson ratios of particle *z* and *y*, respectively; $R^*$ was the equivalent radius, and $R^* = \left(1/R_{za} + 1/R_{yb}\right)^{-1}$ with $R_{za}$ and $R_{yb}$ being the radii of partial sphere *a* and *b*, respectively; $\delta_{zap}$ was the normal overlap; $\beta = \ln e / \sqrt{\ln^2 e * + \pi^2}$ with $e^*$ being the coefficient of restitution; $S_n = 2E^*\sqrt{R^*\delta_{zap}}$; $m^*$ was the equivalent mass, and $m^* = \left(1/m_{za} + 1/m_{yb}\right)^{-1}$ with $m_{za}$ and $m_{yb}$ being the masses of partial sphere *a* and *b*, respectively; $\hat{\text{n}}_{zap}$ was the normal unit contact vector which pointed from the contact point *p* to the center of partial sphere *a*, and $\hat{\text{n}}_{zap} = \left(x_{za} - x_{zap}\right) / \left|x_{za} - x_{zap}\right|$ with $x_{za}$ being the position vector of the center of partial sphere *a*, and $x_{zap}$ being the position vector of the contact point *p*; $S_t = 8G^*\sqrt{R^*\delta_{zap}}$, $G^*$ was the equivalent shear modulus; $\xi_{zap}$ was the total tangential displacement of partial

sphere $a$; $v^n_{zap}$ and $v^t_{zap}$ were the relative normal and tangential velocities of partial sphere $a$ at the contact point $p$; $\mu_s$ was the coefficient of static friction.

The forces and moments acting on multi-sphere particle $z$ were, respectively, expressed as:

$$\mathrm{F}_z = \sum_{a=1}^{N} \sum_{p=1}^{P} \left( \mathrm{F}^n_{zap} + \mathrm{F}^t_{zap} \right) \tag{11}$$

and

$$\mathrm{T}_z = \sum_{a=1}^{N} \sum_{p=1}^{P} \left[ (\mathrm{x}_{zap} - \mathrm{x}_z) \times \mathrm{F}^t_{zap} - \mu_r \left| \mathrm{F}^n_{zap} \right| \left| \mathrm{x}_{zap} - \mathrm{x}_z \right| \hat{\omega}_z \right] \tag{12}$$

where $N$ and $P$ were the numbers of partial spheres and contact points, respectively; $x_z$ was the position vector of the center of particle $z$; $\mu_r$ was the coefficient of rolling friction; $\hat{\omega}_z$ was the unit angular velocity of multi-sphere particle $z$, as shown in Figure 15 [11].

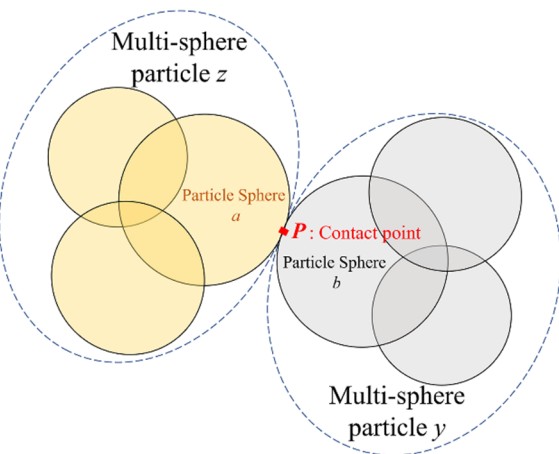

**Figure 15.** Contact force model of multi-sphere particles.

## 5. Determination of Simulation Parameters

The relative simulation parameters are needed when the simulation analysis is performed using DEM. Some of these parameters can be obtained from experiments; see the details in Sections 2 and 3. However, the Poisson's ratio and the coefficients of static friction and rolling friction between the *Cyperus esculentus* seeds are difficult to measure through experiments. Thus, the direct shear test [23] is adopted for calibration. Furthermore, to obtain more accurate simulation parameters, in this section, the PB test was first used to determine the significant performance of each simulation parameter on the simulation results, and then the final value of the simulation parameters was determined combined with the results of the path of steepest ascent method.

### 5.1. The Direct Shear Test

A ZJ strain-controlled direct shear apparatus was used for the direct shear test, as shown in Figure 16. The inner diameter of the direct box was 61.8 mm, and the height was 20 mm. The shear box was evenly filled with *Cyperus esculentus* seeds, and the surface was smoothed. The weight of each test sample was recorded. Each sample was loaded to different normal compressive stresses (50 kPa, 100 kPa, 150 kPa, and 200 kPa), and each test was repeated three times. The results are listed in Table 6.

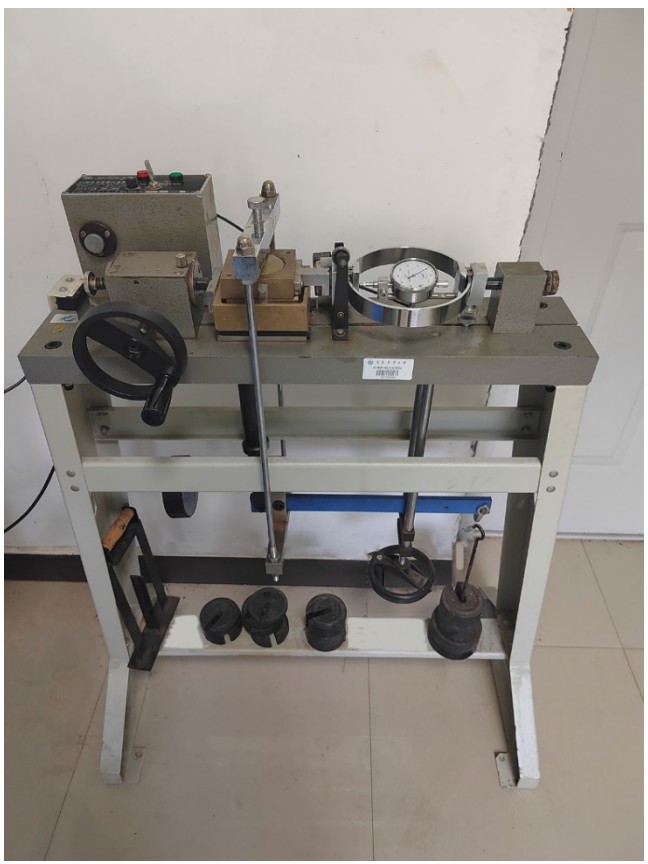

**Figure 16.** ZJ strain-controlled direct shear apparatus.

**Table 6.** Results of the direct shear tests for different varieties of *Cyperus esculentus* seeds.

| Variety | Maximum Shear Strength /kPa | Internal Friction Angle /° | Cohesive Force /kPa |
|---|---|---|---|
| Jinong 1 | 118.23 | 27.03 | 14.11 |
| Jinong 2 | 136.67 | 31.43 | 6.88 |
| Jinong 3 | 105.76 | 25.83 | 14.49 |

*5.2. Plackett–Burman Test and Path of Steepest Ascent Method*

Taking the *Cyperus esculentus* seeds of Jinong 1 as an example, the significance analysis of the simulation parameters of the 9-sphere model for *Cyperus esculentus* was analyzed by the PB test through the direct shear test. On this basis, the value of the simulation parameter was finally determined by the path of steepest ascent method. The shear modulus, static friction coefficient, and restitution coefficient between seed–copper plate and seed–seed were measured by the physical and mechanical property tests in Sections 2 and 3. On this basis, the value range of the factor in the PB test was clarified. For the reference values of Poisson's ratio, the coefficients of rolling friction and static friction between seed–seed were obtained from the standard of the American Society of Agricultural Engineers. Thus, Poisson's ratio was 0.3–0.5, the rolling friction coefficient was 0–0.1, and the static friction coefficient between *Cyperus esculentus* seeds was 0.15–0.55. Then, eight factors were selected as parametric variables, and the other three factors were kept in reserve as dummy variables for error analysis. The maximum shear strength was taken as the response value, two levels of each variable were taken as high (+1) and low (−1), and the factors and levels for simulation are listed in Table 7. Twelve groups of simulations were carried out according to the experimental arrangement. Each case was repeated three times, and the mean value

of three experiments was used as the maximum shear strength in each case. The results are listed in Table 8.

**Table 7.** Factors and levels of the Plackett–Burman test.

| Symbol | Factor | Low Level (−1) | High Level (+1) |
|---|---|---|---|
| A | Poisson's ratio of seed | 0.3 | 0.5 |
| B | Shear modulus of seed/MPa | 30 | 300 |
| C | Coefficient of static friction of seed–seed | 0.15 | 0.55 |
| D | Coefficient of static friction of seed–polymethyl methacrylate | 0.2 | 0.6 |
| E | Coefficient of rolling friction of seed–seed | 0 | 0.1 |
| F | Coefficient of rolling friction of seed–polymethyl methacrylate | 0 | 0.1 |
| G | Restitution coefficient of seed–seed | 0.15 | 0.75 |
| H | Restitution coefficient of seed–polymethyl methacrylate | 0.2 | 0.8 |
| I1, I2, I3 | Virtual parameters | — | — |

**Table 8.** Arrangement and results of the Plackett–Burman test.

| No. | A | B | C | D | E | F | G | H | I1 | I2 | I3 | Y Maximum Shear Strength (kPa) |
|---|---|---|---|---|---|---|---|---|---|---|---|---|
| 1 | 0.5 | 300 | 0.15 | 0.6 | 0.1 | 0.1 | 0.15 | 0.2 | −1 | 1 | −1 | 97.9953 |
| 2 | 0.3 | 300 | 0.55 | 0.6 | 0 | 0 | 0.15 | 0.8 | −1 | 1 | 1 | 127.114 |
| 3 | 0.3 | 30 | 0.15 | 0.2 | 0 | 0 | 0.15 | 0.2 | −1 | −1 | −1 | 51.6994 |
| 4 | 0.3 | 300 | 0.55 | 0.2 | 0.1 | 0.1 | 0.75 | 0.2 | −1 | −1 | 1 | 88.6227 |
| 5 | 0.5 | 30 | 0.55 | 0.6 | 0 | 0.1 | 0.75 | 0.8 | −1 | −1 | −1 | 99.1627 |
| 6 | 0.3 | 30 | 0.55 | 0.2 | 0.1 | 0.1 | 0.15 | 0.8 | 1 | 1 | −1 | 77.4156 |
| 7 | 0.5 | 30 | 0.15 | 0.2 | 0.1 | 0 | 0.75 | 0.8 | −1 | 1 | 1 | 46.1292 |
| 8 | 0.3 | 300 | 0.15 | 0.6 | 0.1 | 0 | 0.75 | 0.8 | 1 | −1 | −1 | 61.1053 |
| 9 | 0.5 | 300 | 0.15 | 0.2 | 0 | 0.1 | 0.15 | 0.8 | 1 | −1 | 1 | 80.4509 |
| 10 | 0.5 | 30 | 0.55 | 0.6 | 0.1 | 0 | 0.15 | 0.2 | 1 | −1 | 1 | 194.423 |
| 11 | 0.5 | 300 | 0.55 | 0.2 | 0 | 0 | 0.75 | 0.2 | 1 | 1 | −1 | 122.911 |
| 12 | 0.3 | 30 | 0.15 | 0.6 | 0 | 0.1 | 0.75 | 0.2 | 1 | 1 | 1 | 81.3848 |

When the processes of the direct shear test were simulated, the Hertz–Mindlin (no-slip) contact model was used, and the EDEM version was EDEM 2018, DEM Solutions, Edinburgh, UK, 2002. To ensure the convergence and stability of the numerical calculation in the simulation, the time step in the simulations was $5 \times 10^{-7}$ s. The simulation time was 12 s. The *Cyperus esculentus* seed particles were generated by a normal distribution in terms of volume, and the sample mass generated was the same as the test mass for each case. At 2 s in the simulation, the plate was subjected to a normal force of 200 kPa in the +$z$ direction and was stabilized for 0.5 s; at 2.5 s in the simulation, the down box started to move at a speed of 0.002 m/s in the +$x$ direction until the end of the simulation, and the screenshots of the simulation for different times are shown in Figure 17a–d.

The maximum shear strength of the seeds was obtained after each simulation, and then, ANOVA was carried out. The results are listed in Table 9. The significance of the $p$ value determines the influence level of the factor on the test index. As a result, it was found that the static friction coefficient of seed–seed had a significant influence on the maximum shear strength.

The results of the PB test combined with ANOVA can only determine which parameter is significant, but it is difficult to effectively calibrate the static friction coefficient of seed–seed. Therefore, the path of steepest ascent method is needed, which can quickly approach the optimum value of the critical factors. The Pareto chart of PB showed that the static friction coefficient of seed–seed had a positive effect on the response value in the PB test, as shown in Figure 18. Therefore, the factor became the rising path in the path of steepest ascent test.

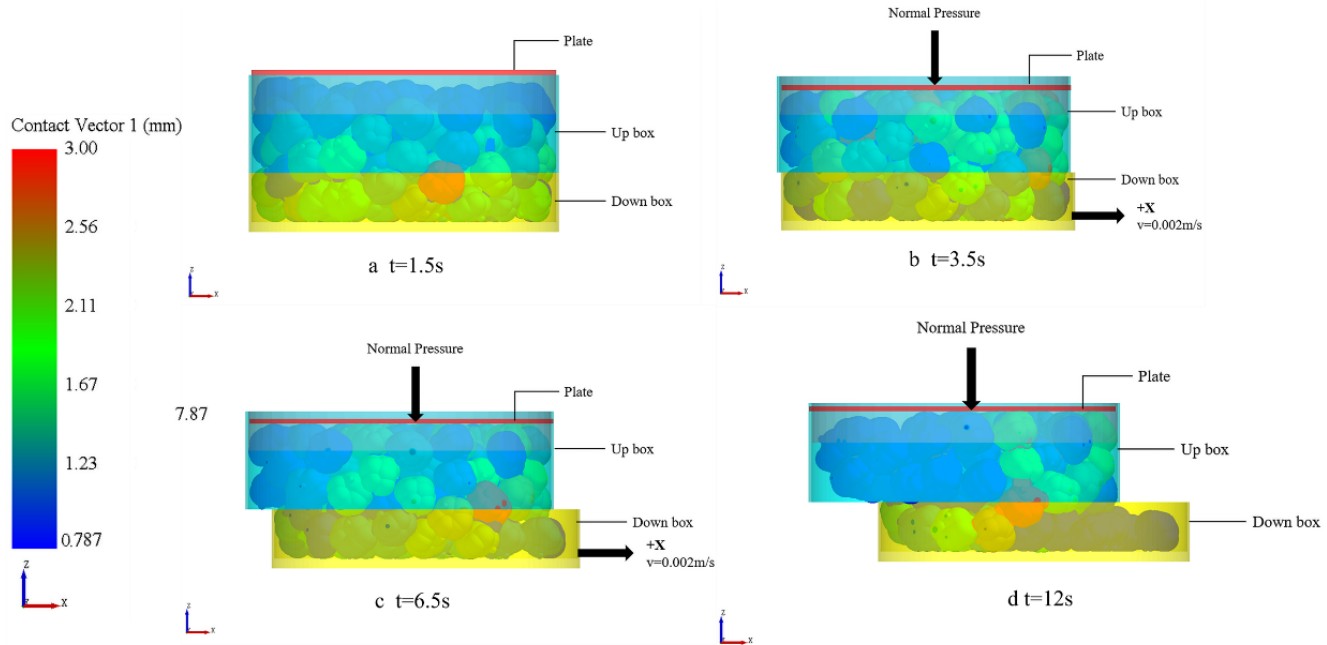

**Figure 17.** Snapshots of the simulation of the direct shear test of Jinong 1 seeds at different times using the 9-sphere model: (**a**) $t$ = 1.5 s; (**b**) $t$ = 3.5 s; (**c**) $t$ = 6.5 s; and (**d**) $t$ = 12 s.

**Table 9.** Significant analysis of factors in the Plackett–Burman test.

| Factor | Sum of Squares | F Value | p Value | Significance |
|--------|---------------|---------|---------|--------------|
| A | 1969.42 | 3.01 | 0.1813 | 3 |
| B | 65.26 | 0.100 | 0.7729 | 7 |
| C | 7051.12 | 10.77 | 0.0464 | 1 |
| D | 3134.91 | 4.79 | 0.1165 | 2 |
| E | 0.73 | $1.121 \times 10^{-3}$ | 0.9754 | 8 |
| F | 511.55 | 0.78 | 0.4419 | 6 |
| G | 1403.62 | 2.14 | 0.2394 | 5 |
| H | 1768.04 | 2.70 | 0.1989 | 4 |

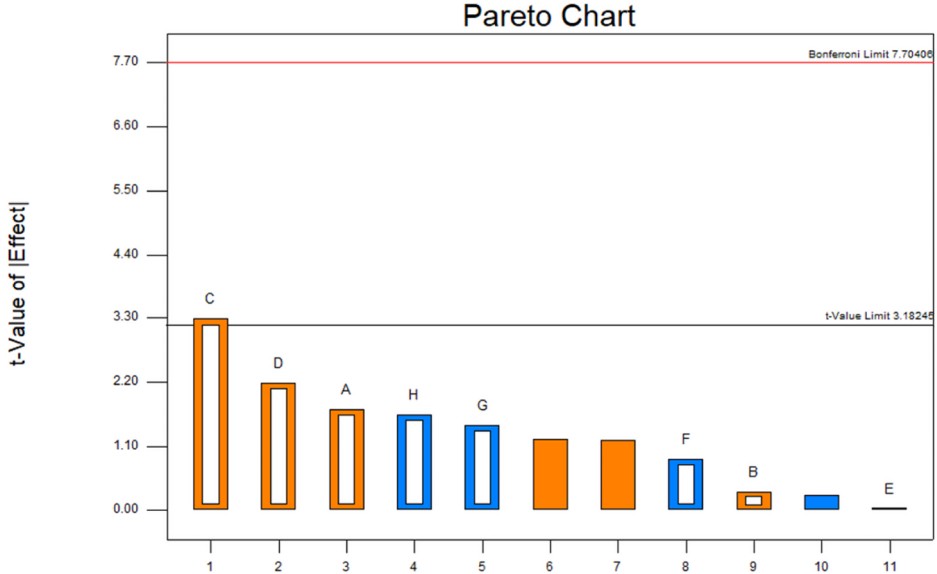

**Figure 18.** Pareto chart of the PB test.

Moreover, the arrangement of the path of steepest ascent test is determined according to the value range of the static friction coefficient of seed–seed in the PB test. The arrangement and results of the path of steepest ascent test are listed in Table 10.

**Table 10.** Arrangement and results of the path of steepest ascent method.

| No. | Coefficient of Static Friction of Seed–Seed | Maximum Shear Strength /kPa | Relative Error |
|---|---|---|---|
| 1 | 0.15 | 50.73 | 57.09% |
| 2 | 0.25 | 62.54 | 47.10% |
| 3 | 0.35 | 95.80 | 18.98% |
| 4 | 0.45 | 108.67 | 8.09% |
| 5 | 0.55 | 122.88 | 3.93% |

As a result, with the increase in the static friction coefficient of seed–seed, the value of the maximum shear strength in the simulation increased. The smallest relative error was observed in test No. 5, which indicated that the optimal value of the factor was close to a value of 0.55. Therefore, the static friction coefficient of seed–seed was taken as 0.55. The static friction coefficients of seed–seed of Jinong 2 and Jinong 3 were also calibrated in the same way, and a value of 0.35 was appropriate for both. In addition, the value of the other simulation parameters was taken as the value of the 0-level in the PB test.

## 6. Analysis and Validation

### 6.1. Piling Tests

A cube container made of polymethyl methacrylate was used for the piling test [24] and had a length, width, and height of 120 mm. The thickness was 5 mm. The container was without a lid. First, the *Cyperus esculentus* seeds were poured into the container, and the upper surface of the seeds was flattened with a scraper. Second, the right baffle was pulled out at a speed of 1 m/s. The *Cyperus esculentus* seeds began to fall and pile until they were stable, and then the static angle of repose was formed, as shown in Figure 19a. To improve the measurement accuracy of the static angle of repose, the collected images were binarized, as shown in Figure 19b. The test of each variety was repeated five times, and the mean value of the five experiments was used as the value for the static angle of repose in each case.

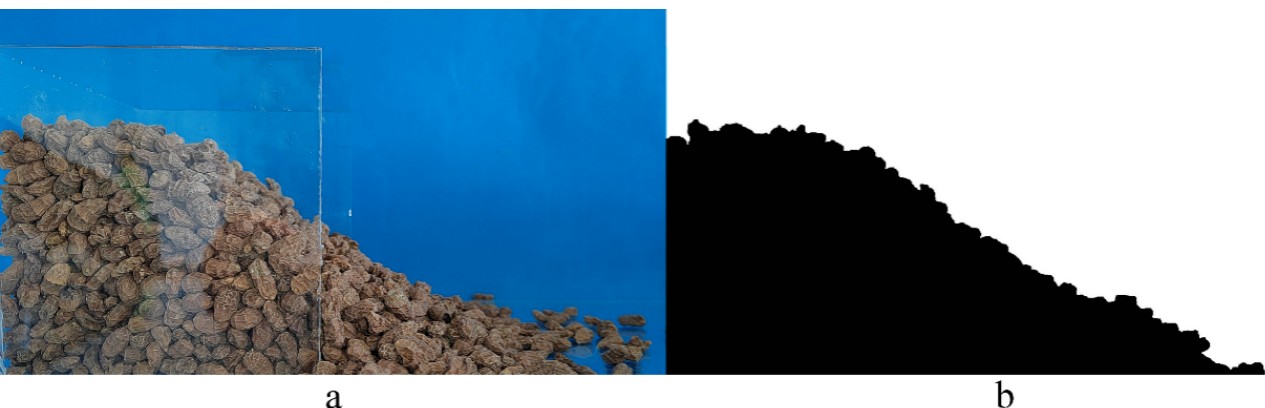

**Figure 19.** Packing test: (**a**) photograph of the piling test captured by a high-speed camera; (**b**) image binarization.

### 6.2. Bulk Density Tests

A box made of polymethyl methacrylate was used for the bulk density test and had a length, width, and height of 75 mm, 75 mm, and 65 mm. The *Cyperus esculentus* seeds were released above the box. The gravity acceleration caused the seed particles to fall into the box and a conical pile to form at the top of the box. When the seeds were stable, a scraper

made of polymethyl methacrylate was used to scrape the redundant seeds with a speed of 0.5 m/s, as shown in Figure 20a–c. The bulk density was calculated according to the ratio of the seeds' mass left in the box to the volume of the box. The experimental data for each condition were repeated eight times, and the mean value was used as the value for bulk density in each case.

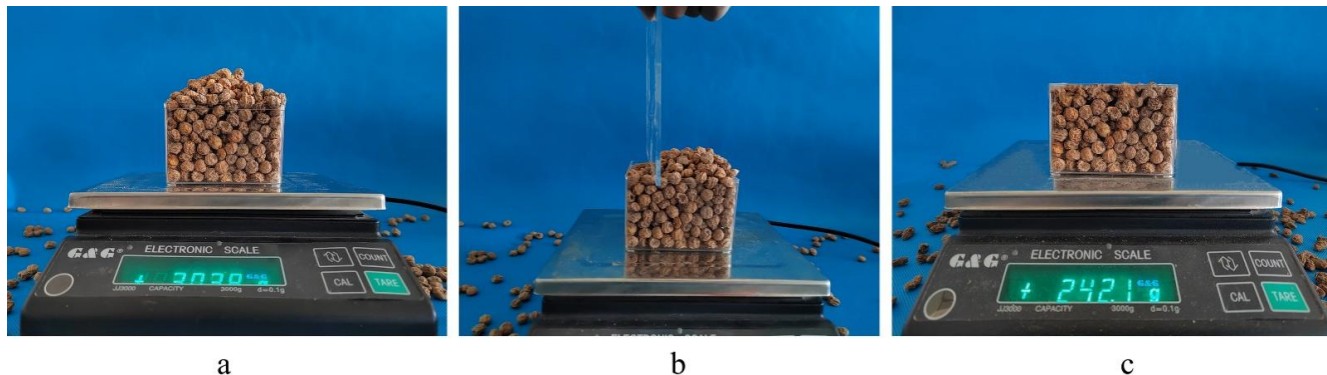

**Figure 20.** (**a**–**c**) Bulk density test of the *Cyperus esculentus* seeds.

*6.3. Simulation Analysis*

In this paper, the piling test and the bulk density test were both simulated for the seeds of the three varieties of *Cyperus esculentus*. Particle models with different filing spheres were used in the simulation. The simulations were carried out using the 7-sphere, 9-sphere, and 11-sphere models for Jinong 1, the 9-sphere, 11-sphere, and 13-sphere models for Jinong 2, and the 7-sphere, 9-sphere, and 11-sphere models for Jinong 3. The simulation parameters of polymethyl methacrylate in the simulation were obtained from Ref. [25]. The simulation parameters are listed in Table 11.

**Table 11.** Simulation parameters of the piling test.

| Parameter | Jinong 1 | Jinong 2 | Jinong 3 | Polymethyl Methacrylate |
|---|---|---|---|---|
| Poisson's ratio | 0.4 | 0.4 | 0.4 | 0.32 |
| Density kg/m$^3$ | 1340 | 1270 | 1190 | 1190 |
| Shear modulus MPa | 165 | 165 | 165 | 1197 |
| Coefficient of restitution | 0.45 | 0.45 | 0.45 | 0.55 |
| Coefficient of static friction | 0.55 | 0.35 | 0.35 | 0.34 |
| Coefficient of rolling friction | 0.05 | 0.05 | 0.05 | 0.05 |

The Hertz–Mindlin (no-slip) contact model was used. To ensure the convergence and stability of the numerical calculation in the simulation, the time step in the simulations was $5 \times 10^{-7}$ s.

In the piling test, the simulation time was 4 s. The simulation was performed in two stages. In the first stage, the *Cyperus esculentus* seed particles were generated by a normal distribution in terms of volume, and the total mass produced in the simulation was the same as that in the test. The volume, mass, and moment of inertia of a single particle for the three varieties are listed in Table 12, which are guaranteed to not change with the increase in the number of filing spheres.

**Table 12.** Generation of particle parameters by simulation.

| Variety | Volume *1/m³ $V$ | Moment of inertia /kg·m² $I_x$ *2 | $I_y$ *3 | $I_z$ *4 |
|---|---|---|---|---|
| Jinong 1 | $2.91722 \times 10^{-6}$ | $1.37398 \times 10^{-10}$ | $1.21829 \times 10^{-10}$ | $1.14194 \times 10^{-10}$ |
| Jinong 2 | $2.66842 \times 10^{-6}$ | $1.70719 \times 10^{-10}$ | $6.64186 \times 10^{-11}$ | $1.49946 \times 10^{-10}$ |
| Jinong 3 | $5.39506 \times 10^{-6}$ | $3.49005 \times 10^{-10}$ | $2.96202 \times 10^{-10}$ | $2.83974 \times 10^{-10}$ |

*1 $V = 4/3\pi abc$; $m = \rho V$; *2 $I_x = 1/5m(b^2 + c^2)$; *3 $I_y = 1/5m(a^2 + c^2)$; *4 $I_z = 1/5m(a^2 + b^2)$; where a, b, and c are half of the length, thickness, and width of the seed.

The second stage of the simulation started after 0.5 s for particle assembly stabilization. The right baffle began to move upward at a speed of 1 m/s, and the *Cyperus esculentus* seed particles slid out until a stable static angle of repose was formed. A screenshot of the simulation is shown in Figure 21.

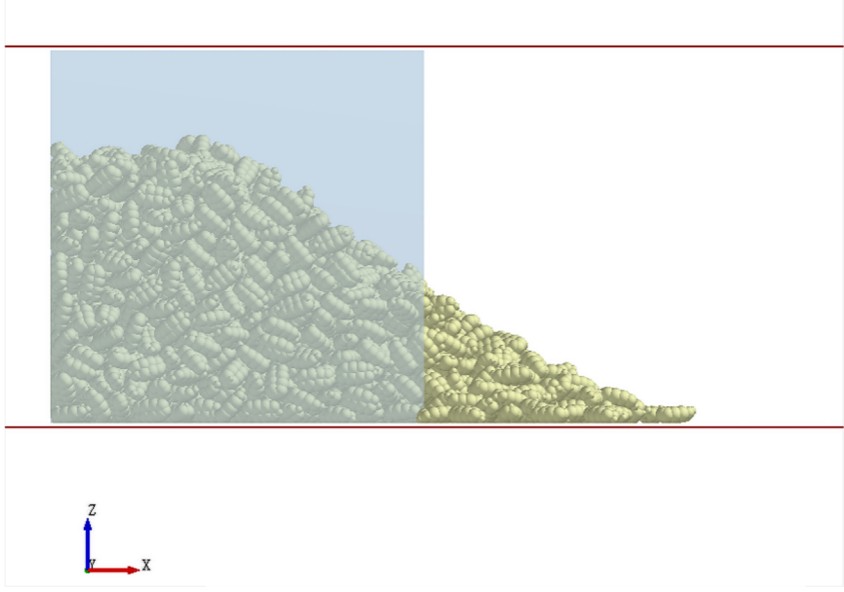

**Figure 21.** Snapshot of the simulation of the piling test of Jinong 2 seeds using the 11-sphere model.

The comparisons between the experimental results and the simulated results of static angle of repose are shown in Figure 22a–c. With the increase in the number of filing spheres, the simulated results were consistent with those obtained experimentally in the piling test. Except for the 7-sphere of Jinong 1 and the 9-sphere of Jinong 2, the mean value of the simulated results fluctuated within the standard deviation of the experimental results, and the relative errors between the simulated results and the experimental results in terms of the static angle of repose for the other particle models were in the range from 0.25% to 2.67%. The mean values of the simulated results were all within the margin of the standard errors of the experimental results.

In the bulk density test, the simulation time was 2 s. The simulation was performed in two stages. In the first stage, the *Cyperus esculentus* seed particles were generated by a normal distribution in terms of volume. The volume, mass, and moment of inertia of a single particle for the three varieties were still the same as the ones listed in Table 12, which are guaranteed to not change with the increase in the number of filing spheres.

The second stage of the simulation started after 0.5 s for particle assembly stabilization. The scraper began to move at a speed of 0.5 m/s and was used to scrape the redundant seeds. The particles' mass left inside the box can be calculated, thus, the bulk density of the particles was calculated. Every numerical experiment was repeated eight times. The bulk density test simulation process for the 9-sphere model of Jinong 3 is shown in Figure 23a–c.

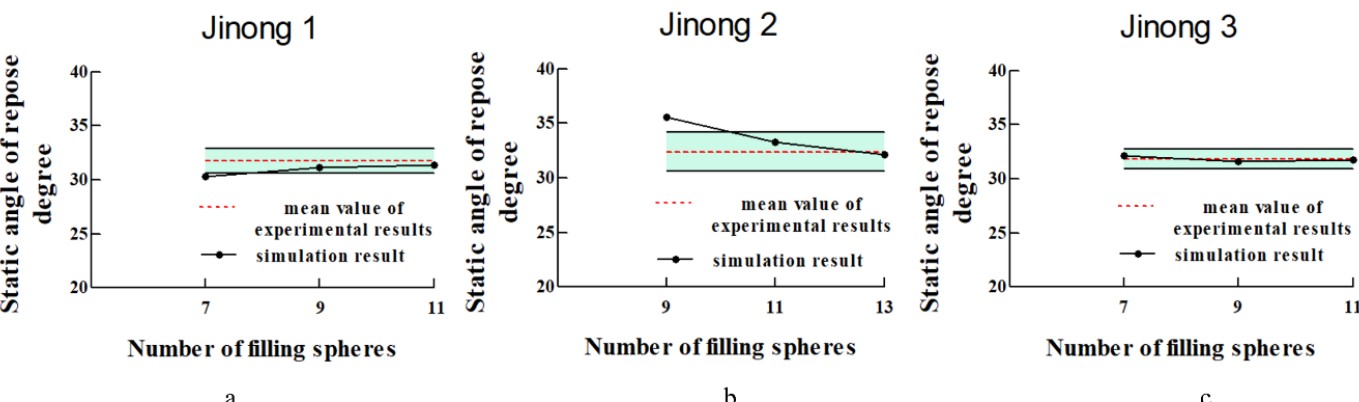

**Figure 22.** (**a–c**) Variations of the simulated static angle of repose versus the number of filing spheres for different varieties of seeds.

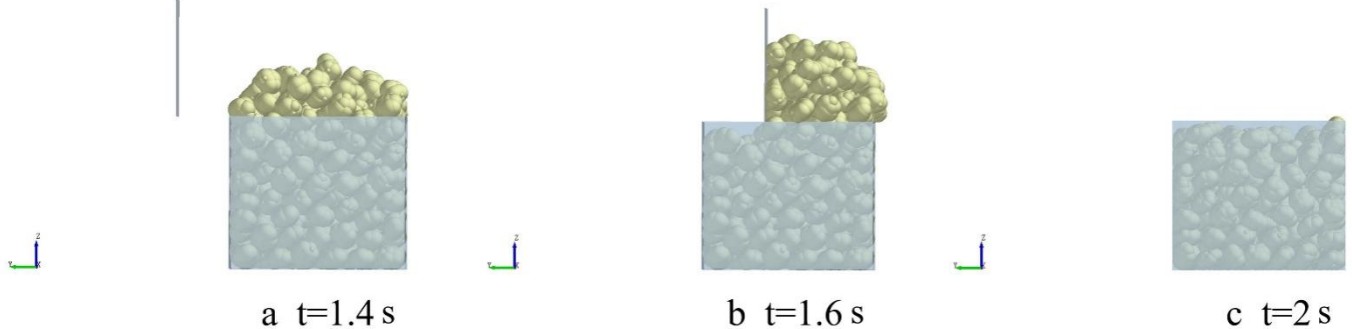

a  t=1.4 s                    b  t=1.6 s                    c  t=2 s

**Figure 23.** Snapshots of the simulation of the bulk density test of Jinong 3 seeds at different times using the 9-sphere model: (**a**) *t* = 1.4 s; (**b**) *t* = 1.6 s; (**c**) *t* = 2 s.

The comparisons between the experimental results and the simulated results of bulk density test are shown in Figure 24a–c. The conclusion was consistent with that obtained in the simulation of the piling test. Except for the 9-sphere of Jinong 2, the mean value of the simulated results fluctuated within the standard deviation of the experimental results. The mean values of the simulated results were all within the margin of the standard errors of the experimental ones. Thus, the feasibility and rationality of the *Cyperus esculentus* seed models were established and the parameter selections in this paper were further verified.

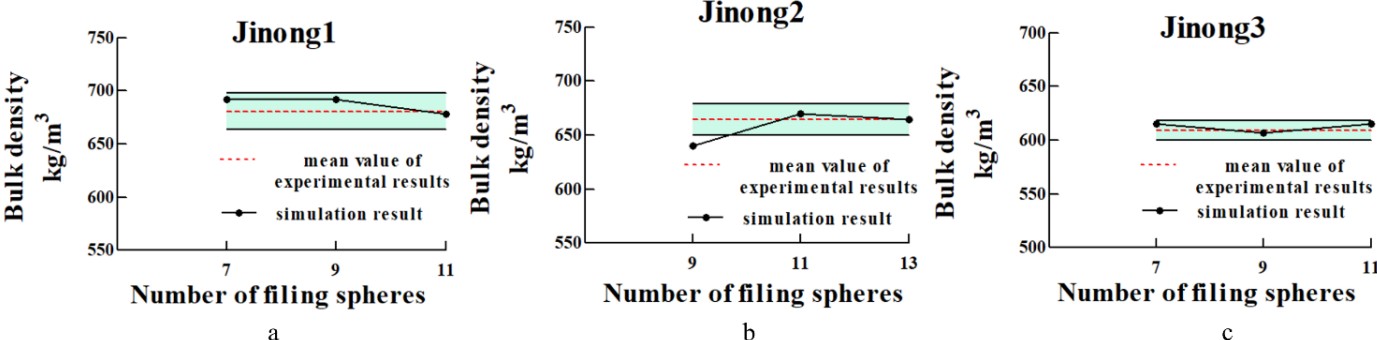

**Figure 24.** (**a–c**) Variations of the simulated bulk density versus the number of filing spheres for different varieties of seeds.

## 7. Conclusions

In this paper, three varieties of *Cyperus esculentus* seeds with irregular shapes (Jinong 1, Jinong 2, and Jinong 3) were used to study their geometrical shapes, and the sizes of the

seeds were measured and analyzed. The point cloud data of the outlines were obtained by 3D scanning technology. The modeling methods for a single *Cyperus esculentus* seed particle and for a *Cyperus esculentus* seed particle assembly were proposed. Moreover, the physical and mechanical properties of the seeds from three varieties of *Cyperus esculentus* were studied. Partial physical and mechanical parameters were obtained by experiments. The PB test and path of steepest ascent method were both adopted to correct and calibrate the simulation parameters, which were difficult to obtain through experiments, and simulation of the direct shear test was used for calibration. Finally, by comparing the simulated results and experimental results in the piling tests and the bulk density test, the simulated results were close to those obtained experimentally. Therefore, the feasibility and validity of the modeling method for the *Cyperus esculentus* seed particles that we proposed and the simulation parameters that were obtained were verified. The following conclusions are based on the data obtained in the current study:

(1) The sizes of the *Cyperus esculentus* seed particles all had a normal distribution, and a certain functional relationship was identified between the primary dimension and other secondary dimensions. The width of the seed was the primary dimension, and the other secondary dimensions (length and thickness) were calculated based on their relationships with the primary dimension. On this basis, an approach for modeling *Cyperus esculentus* seed particles based on the MS method was proposed. The 7-sphere, 9-sphere, and 11-sphere models were constructed for the seeds of Jinong 1 and Jinong 3, and the 9-sphere, 11-sphere, and 13-sphere models were constructed for the seeds of Jinong 2;

(2) The mechanical properties of the *Cyperus esculentus* seeds were tested and analyzed. The elastic modulus of the seed, the restitution coefficient between seed–seed, the restitution coefficient between the seed and the contact material, and the static friction coefficient were all obtained through experiments. Thus, the value range of the simulation parameters was determined, and then, significance analysis of the simulation parameters was carried out by using the PB test design method through the direct shear test in the simulation. It was found that the static friction coefficient between seed–seed had the most significant effect on the results. On this basis, the value of the simulation parameters was further confirmed through the path of steepest ascent method;

(3) The piling tests and the bulk density test were both adopted for further modeling verification. With the increase in the number of filing spheres, the simulated results were consistent with those obtained experimentally in the piling test and the bulk density test. Except for the 7-sphere of Jinong 1 and 9-sphere of Jinong 2 in the piling test, and the 9-sphere of Jinong 2 in the bulk density test, the mean value of the simulated results fluctuated within the standard deviation of the experimental results. The mean values of the simulated results were all within the margin of the standard errors of the experimental results. Thus, the feasibility and rationality of the *Cyperus esculentus* seed models established and the parameters' selection in this paper were further verified;

(4) Future research will be conducted as follows: The established seed model and the simulation parameters selected will be applied to the analysis of the working process of the seed metering device and the cleaning apparatus of the *Cyperus esculentus* seeds in simulations. In addition, other types of irregular seed modeling will be studied to enrich the theory of irregular seed modeling.

**Author Contributions:** Conceptualization, T.X. and J.W.; methodology, T.X. and R.Z.; software, T.X. and Y.W.; validation, W.F., F.Z. and R.Z.; formal analysis, T.X.; investigation, R.Z.; resources, W.F.; data curation, Y.W. and R.Z.; writing—original draft preparation, T.X.; writing—review and editing, T.X.; visualization, F.Z.; supervision, J.W.; project administration, J.W.; funding acquisition, F.Z. and J.W. All authors have read and agreed to the published version of the manuscript.

**Funding:** This research was funded by the National Key Research and Development Program of the 13th Five-Year Plan, No. 2019YFD1002602; Provincial Major Science and Technology Project, No. 20200502006NC.

**Institutional Review Board Statement:** The study did not require ethical approval.

**Informed Consent Statement:** Not applicable.

**Data Availability Statement:** The study did not report any data.

**Acknowledgments:** In this paper, we received technical support from the College of Biological and Agricultural Engineering in Jilin University, including the licensed software of EDEM.

**Conflicts of Interest:** The authors declare no conflict of interest.

## Nomenclature

| | |
|---|---|
| DEM | Discrete element method |
| MS | Multisphere method |
| PB | Plackett–Burman |
| ANOVA | Analysis of Variance |
| $L$ | Length, mm |
| $W$ | Width, mm |
| $T$ | Thickness, mm |
| $E^*$ | Young's modulus, MPa |
| $F$ | Normal force to the seed, N |
| $\mu$ | Poisson's ratio, dimensionless |
| $D$ | Deformation of the seed, mm |
| $R$ | Minimum curvature radii of the seeds with the compression probe and undersurface, mm |
| $R'$ | Maximum curvature radii of the seeds with the compression probe and undersurface, mm |
| $K_U$ | Constant |
| $H'$ | Thickness of the seed when compressed, mm |
| $L'$ | Length of the seed when compressed, mm |
| $G^*$ | Shear modulus, MPa |
| $e^*$ | Coefficient of restitution, dimensionless |
| $H$ | Rebound height of the seed, mm |
| $H$ | Release height of the seed, mm |
| $v_0$ | Release velocity of seed No. 1, m/s |
| $v_1$ | Velocity of seed No. 2 after collision, m/s |
| $v_2$ | Velocity of seed No. 1 after collision, m/s |
| $h_1$ | Rebound height of seed No. 2 after collision, mm |
| $h_2$ | Rebound height of seed No. 1 after collision, mm |
| $G$ | Gravitational acceleration, m/s$^2$ |
| $m$ | Mass, g |
| $\theta$ | Angle between the inclined apparatus and horizontal plane, ° |
| $\mu_s$ | Coefficient of static friction, dimensionless |
| $E_z, E_y$ | Young's moduli of particle $z$ and $y$, MPa |
| $\mu_z, \mu_y$ | Poisson ratios of particle $z$ and $y$, dimensionless |
| $R^*$ | Equivalent radius, mm |
| $R_{za}, R_{yb}$ | Radii of elemental sphere $a$ and $b$, mm |
| $\delta_{zap}$ | Normal overlap |
| $m^*$ | Equivalent mass, g |
| $m_{za}, m_{yb}$ | Masses of elemental sphere $a$ and $b$, g |
| $x_{za}, x_{zap}$ | Position vectors of the center of elemental sphere $a$ and the contact point $p$ |
| $\varsigma_{zap}$ | Total tangential displacement of elemental sphere $a$, mm |
| $v^n{}_{zap}, v^t{}_{zap}$ | Relative normal and tangential velocities of elemental sphere $a$ at the contact point $p$, m/s |
| $N, P$ | Numbers of elemental spheres and contact points |
| $x_z$ | Position vector of the center of particle $z$ |
| $\mu_r$ | Coefficient of rolling friction, dimensionless |
| $\hat{\omega}_z$ | Unit angular velocity of particle $z$, rad/s |
| $V$ | Volume, m$^3$ |
| $Ix, Iy, Iz$ | Moment of inertia, kg·m$^2$ |
| $a$ | Half of the length of the seed, mm |
| $b$ | Half of the thickness of the seed, mm |
| $c$ | Half of the width of the seed, mm |

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
