# Peer review of "A DEM-Based Modeling Method and Simulation Parameter Selection for Cyperus esculentus Seeds"

_processes, doi:10.3390/pr10091729_

Round 1

Reviewer 1 Report

The authors modeled the shape of Cyperus esculentus seeds and compared the DEM simulations with experiments. The paper is on a topic of importance and will be of interest to others working in the field. I recommend publication with minor changes.

1.    Since the coefficient of static friction of see-polymethyl methacrylate and the restitution coefficient of seed-seed have been measured experimentally (See table 4 and 5), why are they still included in the PB test (table 7)? 

2.    What are the dimensions of the boxes used in the direct shear test? In Figure 16, what is the description of contact vector 1?

3.    What is the cost of changing the model from 7-shperes to 11 spheres? More computational time?

4.    It is not surprised that only static friction is statistically significant in your two simulations since in both cases the particles are dense packed and moving slowly. If you increase the speed in the direct shear case, restitution of coefficient and rolling friction could be more important.

Author Response

Dear Reviewer,

Thank you very much for your advice. We have revised the manuscript, and would like to re-submit it for your consideration. We have addressed the comments and the amendments are highlighted in red in the revised manuscript. Point by point responses to your comments are listed below this letter. We would like to express our sincere thanks to you for the constructive and positive comments.

We hope that the revised version of the manuscript is now acceptable for publication. 

I look forward to hearing from you soon.

With best wishes,

Yours sincerely,

Tianyue Xu

First author

Reviewer 2 Report

This paper calibrates the DEM parameters of Cyperus esculentus seed particles. A modelling method based on the multisphere method is proposed to establish sample models of Cyperus esculentus seed. To obtain more accurate simulation parameters, the restitution coefficient, static friction coefficient, and rolling friction coefficient of ‘‘particle–particle” and ‘‘particle-geometry” are determined by the experiments and Plackett-Burman test. Finally, pilling tests were used for modeling verification,which verified the feasibility and validity of the simulation parameters. The paper is well written, the proposed optimal parameter combination could provide a reference for the reasonable selection of simulation parameters of the Cyperus esculentus seed using the EDEM software. There are some issues need to be addressed before further consideration.

1. In page 1, ABSTRACT, this part is complicated and lacks impact, authors are suggested to summarize the main work of this paper briefly and give the highlights of the article. Maybe this part can be improved!

2. Relevant research background needs to be supplemented in INTRODUCTION. I think the advantages of the discrete element method should be added in this part.

3. Please tell more details about the contact model theory in the part of “Materials and Methods”. At the same time, you need to explain why this contact model is used.

4. Table 2: The unit of the mean size should be added.

5. Line 150-153, please check this paragraph.

6. Please keep the consistency of symbols in the formula, like the Ru in formula(2) and the R in formula(3). Also, the sequence number of the article should be adjusted, such as Line 217-218 and Line 292.

7. Table 7:The Low level and High level ofRestitution coefficient of of seed-polymethyl methacrylate is inconsistent with the Table 8, please keep the consistency of this value.

8. In the part2.3 Modeling method of Cyperus esculentus seeds”, the modelling method of single Cyperus esculentus based on the multi-sphere filling method is proposed. You put a lot of work into this part, please explain the advantages of this methods compared with the automatic filling function within EDEM.

9. Line 346 “The maximum shear strength of the seeds was obtained after each simulation” Please tell more details about the method of obtaining this data.

Author Response

(The authors gave the same response as above.)

Reviewer 3 Report

The authors propose a step-by-step methodology to calibrate the DEM parameters (e.g., static friction coefficient, Young’s modulus, restitution coefficient, and so on) for Cyperus esculentus seed particles of different shape and size. The methodology includes experiments, DEM simulations, and statistical analysis. The clump method was applied to represent the particle shape. So, the influence of the number of spheres comprising the clump on the numerical results was also assessed. Although this topic deserves publication, there are deficits of interpretation and presentation in the manuscript that must be overcome:

1) Please, check out, carefully, the manuscript structure: Table 1, Table 2, Fig. 2, Fig. 3, Fig.4, and so on, belong to the “Results and discussion” section (that, by the way, is missing herein) rather than the “Material and Methods” section.

2) DEM model equations (contact models etc.) are not presented in this manuscript.

3) Since DEM codes usually apply an explicit scheme to integrate the balance force equations over time, the time step must be kept less than a critical value to avoid numerical instabilities. Furthermore, the critical DEM time step depends on the particle properties, e.g., Young’s modulus, shear modulus, etc. (Rayleigh, and Hertz criteria). How was the time step established in this work? Could you provide some information about the hardware configuration used for numerical simulations (processor, number of cores, RAM memory, etc.)?

4) According to Eq. 5, the shear modulus is a function of the Poisson’s ratio. Why were both  varied during the Plackett-Burman test (see Table 7)? Again, how was the Poisson’s ratio measured in order to calculate the elastic modulus (Young’s modulus) in Eq. 1, since the grains’ elastic modulus were presented before the parameters analysis (see Page 7, Lines 167-169). It is very confusing.

5) It is well known that the thickness of the collision material (i.e., copper, steel, and polymethyl methacrylate, herein) has a great impact on the particle coefficient restitution value. Why wasn't it considered herein? (please, see Table 4)

6) In Table 9 it is shown that the coefficient of restitution is not significant, and that the unique significant variable was the static friction coefficient. It is obvious that the seed-seed, and seed-wall coefficients of restitution are not significant in a direct shear test since particles are not colliding with either each other or against the wall. It is also obvious that in this case (i.e., direct shear test) the static friction coefficient is of primary importance. It seems to be redundant.

7) In Table 11, the rolling friction coefficients of Jinong 1, 2, and 3 are the same, i.e., 0.05, even though they have different shapes (Fig. 14). Are they not supposed to have different resistance to roll due to their shapes?

8) The same symbols are used for more than one variable: please check the symbols used for the Poisson’s ratio and for the static friction coefficient; for the release heigh in Eq. 6 and for the seed thickness in Eqs. 3-4; and so on.

9) Please, check out all the citations along the text and in the “References” section (formatting issues):

Example: “Xu [11]” should be replaced by “Xu et al.[11]”

10) Please, write “Cyperus esculentus” in italic.

11) The manuscript needs to be carefully revised. Many typos are presented.

Examples:

> Page 2, Line 69: “…through a pilling test…” should be replaced by “…through a piling test…

> Table 3: Jinong 1, “T=(W-5.9840)…” should be replaced by “T=(W-5.894)…”

> Page 6, Line 149: “…simplified Hertz formula in (1).” should be replaced by “…simplified Hertz formula in Eq. 1”. Please, check out how to cite equations along the text; use “Eq.” instead of “Formula”.

> Eqs. 3-4, please, replace “R” and “R´” by “Ru” and “Ru´”, respectively.

> Page 7, Line 182: “The restitution coefficient between seed and seed…” should be replaced by “The seed-seed restitution coefficient…”

> Page 9, Lines 216-217: “…in Section 2.1”. Is “Section 2.1” correct herein?

> In Fig. 14: Please, replace, for example, “7s” by “7 seeds” since “s” seems to be related to time.

> Pages 12 and 13, Lines 292 and 316: “…in Section 1.1 and Section 1.2.”. “Sections 1.1 and 1.2” do not exist in this manuscript. Please, check this out.

> Table 11: replace “Mpa” by “MPa”; and so on.

12) Please, provide the units for length, width etc. in Table 2 (mean, and standard deviation values), and Figs. 2-4 (x-axis); also, units in Figs. 5-7

13) Please, provide a “Nomenclature” section.

Author Response

(The authors gave the same response as above.)

Reviewer 4 Report

1.       Why are these three types of Cyperus esculentus seeds selected to study? Are there only these three types of Cyperus esculentus seeds? The reason for selecting these three types of Cyperus esculentus seeds should be given.

2.       In Fig. 4, is the red line the probability density function of the size distribution? If it is, the function should be given and how to obtain it?

3.       The young's modulus of the seed is calculated according to the Eq. (1). In Eq. (1), Poisson's ratio of the seed has significant influence on the calculated result. As the Poisson's ratio of the seed is hard to measure, the influence of Poisson's ratio should be evaluated and the consideration about the selected value of Poisson's ratio for determining the modulus should be given.

4.       The verification should be improved. Although the comparison of the static angle of repose between experiments and simulations is good, it is not enough to prove the validity of the modeling method and the simulation parameter selection for simulating mechanical behaviors of Cyperus esculentus seeds, as the static angle of repose is only specific mechanical property and cannot represent other mechanical behaviors. The conclusion which the feasibility and validity of the modeling method were verified needs more evidence to support.

Author Response

(The authors gave the same response as above.)

Round 2

Reviewer 3 Report

Reviewer comments have been considered; the manuscript has been improved correspondingly.